# Robotic Completion Thyroidectomy via the Bilateral Axillo-Breast Approach

**DOI:** 10.3390/jcm10081707

**Published:** 2021-04-15

**Authors:** JungHak Kwak, Su-jin Kim, Zhen Xu, Keunchul Lee, Jong-hyuk Ahn, Hyeong Won Yu, Young Jun Chai, June Young Choi, Kyu Eun Lee

**Affiliations:** 1Department of Surgery, Seoul National University College of Medicine, Seoul 03080, Korea; jhkwak908@gmail.com (J.K.); xuzhen8771@gmail.com (Z.X.); drisaac84@gmail.com (J.-h.A.); kyueunlee@snu.ac.kr (K.E.L.); 2Thyroid Center, Division of Surgery, Seoul National University Cancer Hospital, Seoul 03080, Korea; 3Cancer Research Institute, Seoul National University College of Medicine, Seoul 03080, Korea; 4Medical Big Data Research Center, Institute of Medical and Biological Engineering, Seoul National University, Seoul 03080, Korea; 5Department of Surgery, Seoul National University Bundang Hospital, Seongnam 13620, Korea; curitty@gmail.com (K.L.); hyeongwonyu@gmail.com (H.W.Y.); aznagran@gmail.com (J.Y.C.); 6Department of Surgery, Seoul National University Boramae Medical Center, Seoul 07061, Korea; kevinjoon@naver.com

**Keywords:** bilateral axillo-breast approach, completion thyroidectomy, thyroidectomy, robotic surgical procedures, remote access thyroid surgery

## Abstract

Background: Bilateral axillo-breast approach (BABA) robotic thyroidectomy has been successfully performed for thyroid cancer patients with excellent cosmetic results. Completion thyroidectomy is sometimes necessary after thyroid lobectomy, and whether it has a higher complication rate than the primary operation due to the presence of adhesions remains controversial. The aim of this study was to evaluate surgical outcomes, including operation time and postoperative complications, in patients who underwent BABA robotic completion thyroidectomy. Methods: From Jan 2012 to Aug 2020, 33 consecutive patients underwent BABA robotic completion thyroidectomy for a thyroid malignancy after BABA robotic thyroid lobectomy. The procedures were divided into five steps: (1) robot setting and surgical draping, (2) flap dissection, (3) robot docking, (4) thyroidectomy, and (5) closure. Clinicopathological characteristics, operation time, and postoperative complications were reviewed. Results: The total operation time was shorter for completion thyroidectomy than for the initial operation (164.8 ± 31.7 min vs. 179.8 ± 27.1 min, *p* = 0.043). Among the robotic thyroidectomy steps, the duration of the thyroidectomy step was shorter than that of the initial operation (69.6 ± 20.9 min vs. 83.0 ± 19.5 min, *p* = 0.009. One patient (1/33, 3.0%) needed hematoma evacuation under the flap area immediately after surgery. Three patients (3/33, 9.1%) showed transient hypoparathyroidism, and one patient (1/33, 3.0%) had permanent hypoparathyroidism. Two patients (2/33, 6.1%) showed transient vocal cord palsy and recovered within 3 months following the completion thyroidectomy. There were no cases of open conversion, tracheal injury, flap injury or wound infection. Conclusions: BABA robotic completion thyroidectomy could be performed safely without completion-related complication.

## 1. Introduction

Given the desire to avoid visible anterior neck scars after conventional open thyroidectomy, remote access thyroid surgery has been introduced and developed [1,2,3]. The initial experience with remote access thyroid surgery using endoscopic instruments has several limitations: restrictions in instrument manipulation, unstable camera vision, and loss of stereoscopic depth perception by using two-dimensional visualization [1,2,4]. However, following the introduction of robotic surgery, these limitations of endoscopic surgery have been overcome with the da Vinci surgical system (Intuitive Surgical, Sunnyvale, CA, USA). With the accumulation of surgical experience and advancements in robotic surgical systems, robotic surgery has become popular, with an increased demand for remote access thyroid surgery [5].

Various approaches to remote access thyroid surgery have been developed, including the transaxillary, anterior chest, postauricular, transoral, and bilateral axillo-breast approach (BABA) [5]. Among them, the BABA is one of the most popular techniques for performing remote access thyroid surgery worldwide. Since it provides a symmetric anatomic view to the operators, this approach allows the surgeon to have optimal visualization of crucial structures (e.g., parathyroid glands, recurrent laryngeal nerve, and external branch of the superior laryngeal nerve) and to perform precise procedures in a wide surgical working space, resulting in improved clinical outcomes [6,7].

However, in terms of completion thyroidectomy, generally, there are concerns about increased surgical difficulties and risk of complications, especially tracheal injuries, flap injuries, and bleeding. Because of anatomic structural changes due to fibrosis and adhesions from an initial operation, there might be difficulties in dissecting the skin flap and identifying the trachea and isthmus of the remnant thyroid gland during completion thyroidectomy [8,9,10]. To date, however, no study has demonstrated the surgical safety and clinical outcomes of BABA robotic completion thyroidectomy (BABA RCT).

In this article, we investigated the surgical outcomes of BABA RCTs, including postoperative complications and operation time. Additionally, we divided the BABA RCT procedures into five steps and measured the time required to perform each step to compare the procedural time between the initial and completion operations.

## 2. Materials and Methods

We retrospectively reviewed the medical records of 35 consecutive patients who underwent BABA RCT at Seoul National University Hospital (Seoul, Korea) from Jan 2012 to Aug 2020. Among them, two patients who had undergone initial BABA robotic thyroid lobectomy at another hospital were excluded. Therefore, a total of 33 patients who underwent BABA RCT were included in this study. These patients underwent BABA RCT 5.5 months (range: 3–15 months) after BABA robotic thyroid lobectomy. BABA RCTs were performed using the da Vinci surgical system: the da Vinci S (from 2008 to 2010), da Vinci Si (from 2010 to 2017), and da Vinci Xi (since 2017) (Intuitive Surgical, Sunnyvale, CA, USA). This study was approved by the Institutional Review Board of Seoul National University Hospital (IRB Number 2101-055-1187).

The surgical techniques of BABA robotic thyroidectomy have been described in detail elsewhere [4]. A total of 33 patients used the intraoperative neuromonitoring system during BABA RCT to facilitate identification of the recurrent laryngeal nerve and prevent nerve injury. Jackson-Pratt drain was routinely inserted through the left axilla incision after BABA robotic thyroid surgery and removed before discharge. All patients underwent clinical examination, laryngoscopy, or laryngeal sonography for vocal cord movement evaluation, and ionized calcium, serum calcium, phosphorus, and parathyroid hormone (PTH) assessment to evaluate the function of the vocal cords and parathyroid glands. Other clinical data, including patient sex, age, tumor size, pathologic diagnosis, number of harvested and metastatic central lymph nodes (LNs), and all types of postoperative complications (e.g., flap injury, tracheal injury, hematoma, seroma, estimated blood loss, and length of postoperative hospital stay), were collected. Vocal cord assessments were conducted before and 2 weeks after the operation using laryngoscopy or laryngeal sonography. For patients who had vocal cord hypomobility or palsy, repeated examinations were performed at the follow-up visits. Postoperative parathyroid function was assessed by measuring serum ionized calcium, serum calcium, phosphorus, and PTH levels. Low serum PTH levels (<15 ng/mL) or the need for calcium/vitamin D supplementation to maintain serum calcium concentrations and control hypocalcemic symptoms that recovered within 6 months were defined as transient hypoparathyroidism. When no recovery was observed within 6 months after surgery, the conditions were defined as permanent hypoparathyroidism.

The total operation time was defined as the time from general anesthesia to skin closure. The operation time was recorded for each step of BABA robotic thyroidectomy, which were defined as follows: (1) robot setting and draping, (2) flap dissection (Figure 1A,C), (3) robot docking, (4) thyroidectomy (Figure 1B,D), and (5) closure. The robot setting and draping time was defined as the time from induction of general anesthesia to completion of surgical draping and preparation of the robot surgical system. The flap dissection time was defined as the time from hydrodissection with epinephrine solution to completion of skin flap dissection. The robot docking time was defined as the length of time required to dock the robot armed with instruments. The thyroidectomy time was defined as the length of time needed to complete thyroid resection and/or central LN dissection after robot docking. The closure time was defined as the time from completion of the thyroidectomy step to skin closure. The patient data were statistically analyzed using SPSS software for Windows version 25 (Statistical Software, IBM Corp., Chicago, IL, USA). The difference between groups was analyzed with Student’s t-test. *p* values < 0.05 were considered statistically significant.

## 3. Results

### 3.1. Patient’s Baseline Characteristics and Histopathologic Results

The patients’ baseline characteristics are shown in Table 1. The mean age was 35.2 ± 11.6 years at the time of completion thyroidectomy, and BABA RCTs were performed more frequently in female than male patients (24/33 (72.7%) vs. 9/33 (27.3%), respectively). The mean tumor size at the initial operation was 2.2 ± 1.8 cm. Central LN dissection was performed in 26 patients at the initial BABA thyroid lobectomy (78.8%), and 11 patients (33.3%) underwent central LN dissection when undergoing BABA RCT. The interval between the two operations was 5.5 ± 2.6 months (Table 1). After the initial surgery, the histopathologic results showed papillary carcinoma in 27 patients (81.8%), minimally invasive follicular thyroid carcinoma (miFTC) in three patients (9.1%), and widely invasive follicular thyroid carcinoma (wiFTC) in three patients (9.1%) (Table 2). In the histopathologic results after BABA RCT, most of the patients had no tumor (23/33, 69.7%), eight patients had papillary thyroid microcarcinoma (8/33, 24.2%), one patient had nodular hyperplasia (1/33, 3.0%), and one patient had miFTC (1/33, 3.0%).

### 3.2. Detailed Analysis of Operation Time

The total operation time was significantly shorter for BABA RCT than for the initial BABA robotic thyroid lobectomy (164.8 ± 31.7 vs. 179.8 ± 27.1 min, *p* = 0.043, respectively), and a detailed analysis of the operation time showed that among the BABA RCT steps, the thyroidectomy step was significantly shorter in BABA RCT than for the initial BABA robotic thyroid lobectomy (69.6 ± 20.9 vs. 83.0 ± 19.5 min, respectively, *p* = 0.009) (Table 3). However, the operation times for the other BABA robotic thyroidectomy steps (robot setting and draping, flap dissection, robot docking, and closure) were not significantly different between BABA robotic thyroid lobectomy and BABA RCT.

### 3.3. Postoperative Complications

There were no cases of open conversion, tracheal injury, flap injury, or wound infection. One patient (1/33, 3.0%) had a hematoma beneath the skin flap, underwent hematoma evacuation immediately after surgery, and recovered well with no other complications. Two patients (2/33, 6.1%) had seroma under the skin flap following drain removal, and these were aspirated on the first follow-up visits. Three patients (3/33, 9.1%) showed transient hypoparathyroidism, and one patient (1/33, 3.0%) showed permanent hypoparathyroidism. Two patients (2/33, 6.1%) showed transient vocal cord palsy and recovered within 3 months after BABA RCT (Table 4). No significant difference was found in the estimated blood loss (110.3 ± 57.1 vs. 81.8 ± 57.4 mL, respectively, *p* = 0.121) or length of postoperative hospital stay (2.8 ± 0.5 vs. 3.0 ± 0.8 days, respectively, *p* = 0.142) between the initial BABA robotic thyroid lobectomy and BABA RCT.

## 4. Discussion

To the best of our knowledge, this is the first study to evaluate the surgical safety of BABA RCT and investigate the times of the individual steps of BABA robotic thyroidectomy to estimate the surgical difficulty in performing completion thyroidectomy using BABA after BABA robotic thyroid lobectomy.

Since the initial operation leads to postoperative tissue changes, including tissue inflammation, fibrosis, adhesion, and tissue edema, it might be difficult to identify anatomical landmarks during completion thyroidectomy [10]. Therefore, completion thyroidectomy is generally thought to have a higher risk of postoperative complications, including skin flap or tracheal injury and surgical difficulties, thus requiring a longer operation time than the initial operation.

Although some studies have indicated that the risks of two-stage surgery (initial thyroid lobectomy followed by completion thyroidectomy) are not significantly different from those of initial total thyroidectomy [11,12], completion thyroidectomy is nevertheless technically challenging. Furthermore, since completion thyroidectomy is associated with added costs and patient inconvenience, completion thyroidectomy should be performed carefully [13].

In the present study, there were no flap or tracheal injuries while creating the working space for BABA RCT, which is the major concern for surgeons performing this technique. Additionally, the rates of postoperative complications are comparable with previously reported data on BABA robotic total thyroidectomy (permanent hypoparathyroidism (13/872, 1.5%) and permanent vocal cord palsy (2/872, 0.2%)) [14]. In particular, after we divided the procedures of BABA into five steps and analyzed the duration of each step, the results indicated that thyroidectomy plays a pivotal role in shortening the operation time for BABA RCT compared with initial BABA robotic thyroid lobectomy. Although it is more difficult and therefore takes more time to identify the trachea and isthmus of the remnant thyroid gland in BABA RCT, fewer central LN dissections and no isthmectomy need to be made in the thyroidectomy step, which may explain the shortened operation time. Despite the postoperative changes following the initial surgery, the duration of the flap dissection step for BABA RCT was not significantly different from that for the initial BABA robotic thyroid lobectomy.

With the development and advancement of various approaches for robotic thyroid surgery, many surgeons have questioned the feasibility of completion thyroidectomy via the same robotic approach. Surgeons may encounter severe inflammatory changes or adhesions during RCT, which is also an important issue in conventional completion thyroidectomy. Only a few studies, including case series and case reports, have described the surgical outcomes in patients who have undergone RCT [15,16,17,18]. Razavi et al. (2018) [16] reported one successful case of completion thyroidectomy via the transoral endoscopic thyroidectomy vestibular approach (TOETVA), in which the initial procedure was also performed via TOETVA. According to Stang et al. (2018) [17], among 301 patients who underwent transaxillary robotic thyroidectomy, 20 underwent completion thyroidectomy via the same approach, but this study did not show the detailed surgical outcomes, including complications or operation times, of this latter group of patients.

In the present study, we analyzed a larger number of patients who underwent BABA RCT (*n* = 33) than previously reported for BABA endoscopic completion thyroidectomy and comparatively analyzed the operation times between the initial BABA robotic thyroidectomy and BABA RCT. Kim et al. [19] first reported the surgical outcomes of endoscopic completion thyroidectomy via BABA for 13 patients, describing the surgical outcomes of BABA endoscopic completion thyroidectomy and revealing excellent patient satisfaction with the cosmetic results. [19].

Following the introduction of BABA robotic thyroid surgery, several studies have demonstrated its surgical safety and oncological completeness [20,21,22]. A propensity score matching study reported by Lee et al. [20] found that the percentage of patients with stimulated-Tg levels < 1.0 ng/mL following BABA robotic thyroidectomy was equivalent to that following conventional open thyroidectomy (64.2% and 69.0%, respectively, *p* = 0.593). BABA also provides excellent cosmetic outcomes, requiring only small incisions on both sides of the nipple-areolar and axillary areas and preventing the creation of visible anterior neck scars, thus overcoming the patients’ cosmetic concerns, especially for young women or patients with hypertrophic scar or keloid tendencies [23].

At our institution, completion thyroidectomies have been considered for differentiated thyroid carcinoma patients who are categorized as intermediate or high risk based on American Thyroid Association risk stratification system [11]. In the present study, completion thyroidectomy was performed postoperatively after a mean of 5.5 months (range: 3–15 months) (range: 3–15 months). Generally, completion thyroidectomy is performed within the first week or 3 months after the initial surgery to minimize the risk of complications due to postoperative inflammation and adhesions [24]. However, some studies have reported that the interval from the initial surgery to completion thyroidectomy does not have an effect on the rate of postoperative complications after completion thyroidectomy [25,26]. Additionally, in a systemic review and meta-analysis by Bin et al. [27], delayed completion thyroidectomy (after 90 days) was associated with a lower rate of overall surgical complications than early completion thyroidectomy (7 to 90 days) (OR = 1.55; 95% CI, 1.00–2.42; *z* = 1.95, *p* = 0.05). Therefore, the optimal surgical timing of completion thyroidectomy remains unclear.

This study was a retrospective study, and the presence of a heterogeneous group of patients may influence the incidence of postoperative complications and the operation time. Despite these limitations, this study investigated a relatively large number of consecutive patients who underwent BABA RCT over 8 years at a tertiary referral hospital.

## 5. Conclusions

This study is the first report to evaluate surgical outcomes and perform a detailed analysis of the operation time in patients who underwent BABA RCTs. BABA RCT had a shorter duration than the initial operation without severe postoperative complications. Therefore, this study suggests that the BABA is a safe and feasible procedure for completion thyroidectomy following BABA robotic thyroid lobectomy.

## Figures and Tables

**Figure 1 jcm-10-01707-f001:**
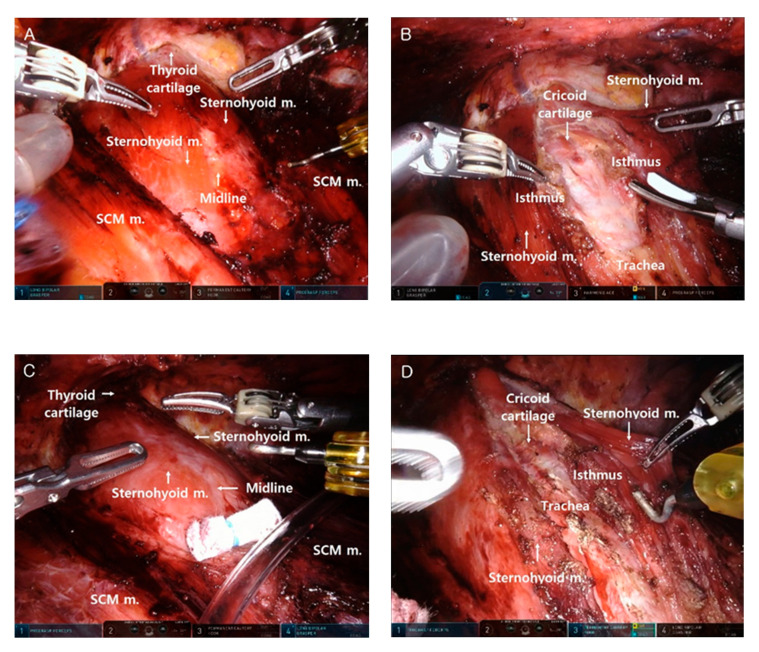
Comparison of surgical field between bilateral axillo-breast approach (BABA) robotic thyroid lobectomy and BABA robotic completion thyroidectomy. (**A**) Complete elevation of the skin flap at BABA robotic thyroid lobectomy. (**B**) Midline division and isthmectomy at BABA robotic thyroid lobectomy. (**C**) Complete elevation of the skin flap at BABA robotic completion thyroidectomy (BABA RCT). (**D**) Midline division and isthmus identification at BABA RCT. SCM: sternocleidomastoid muscle; Sternohyoid m.: sternohyoid muscle.

**Table 1 jcm-10-01707-t001:** Patient baseline characteristics.

Variables	Value
Mean age (years, mean ± SD)	35.2 ± 11.6
Sex	
Male	9 (27.3%)
Female	24 (72.7%)
Body mass index (mean ± SD)	24.5 ± 5.2
Tumor size (initial operation) (mean ± SD)	2.2 ± 1.8
Tumor location (initial operation)	
Right	18 (54.5%)
Left	15 (45.5%)
Central LN dissection (initial operation)	
No	7 (21.2%)
Yes	26 (78.8%)
Central LN dissection (completion operation)	
No	22 (66.7%)
Yes	11 (33.3%)
Interval between the two operations (months, mean ± SD)	5.5 ± 2.6

SD: standard deviation.

**Table 2 jcm-10-01707-t002:** Histopathologic results of initial BABA robotic thyroid lobectomy.

T Stage	Total, *n* = 33
T1	12 (36.4%)
T2	7 (21.2%)
T3	14 (42.4%)
T4	0 (0%)
**N stage**	
Nx	7 (21.2%)
N0	7 (21.2%)
N1a	19 (57.6%)
**Final histopathologic type**	
PTMC	15 (45.5%)
PTC	6 (18.2%)
FVPTC	6 (18.2%)
miFTC	3 (9.1%)
wiFTC	3 (9.1%)

BABA: bilateral axillo-breast approach; PTMC: papillary thyroid microcarcinoma; PTC: papillary thyroid carcinoma; FVPTC: follicular variant papillary thyroid carcinoma; miFTC: minimally invasive follicular thyroid carcinoma; wiFTC: widely invasive follicular thyroid carcinoma.

**Table 3 jcm-10-01707-t003:** Detailed analysis of operation time for each step of BABA robotic thyroidectomy.

	Initial Lobectomy	Completion Thyroidectomy	*p*
Total operation time	179.8 ± 27.1 (min)	164.8 ± 31.7 (min)	*p* = 0.043
Robot setting & draping	22.8 ± 8.6 (min)	20.7 ± 7.3 (min)	*p*= 0.286
Flap dissection	42.2 ±13.9 (min)	43.8 ± 18.8 (min)	*p* = 0.684
Robot docking	7.5 ± 6.6 (min)	8.1 ± 7.8 (min)	*p* = 0.728
Thyroidectomy	83.0 ± 19.5 (min)	69.6 ± 20.9 (min)	*p* = 0.009
Closure	24.3 ± 13.3 (min)	22.5 ± 9.8 (min)	*p* = 0.551

BABA: bilateral axillo-breast approach.

**Table 4 jcm-10-01707-t004:** Postoperative complications following BABA robotic completion thyroidectomy.

Postoperative complications	*n* (%)
Flap injury	0 (0%)
Trachea injury	0 (0%)
Wound infection	0 (0%)
Hematoma	1 (3.0%)
Seroma	2 (6.1%)
Transient hypoparathyroidism	3 (9.1%)
Permanent hypoparathyroidism	1 (3.0%)
Transient recurrent laryngeal nerve palsy	2 (6.1%)
Permanent recurrent laryngeal nerve palsy	0 (0%)

## Data Availability

The data presented in this study are available on request from the corresponding author (S.K.) upon reasonable request.

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
