# Peer review of "Robotic Completion Thyroidectomy via the Bilateral Axillo-Breast Approach"

_jcm, 2021, doi:10.3390/jcm10081707_

Round 1

Reviewer 1 Report

Very interesting work on a particular, modern, current aspect of thyroid surgery: robotic completion. Such a large number of operating cases of a niche intervention (even if in a relatively and rightly long period: January 2012 - August 2020)
underlines the volume of activity and - therefore - the experience and impact on the effectiveness of the working group.
The study, preparation and post-operative evaluatiuos of patients follows the most up-to-date schemes of international literature: clinical examination, laryngoscopy or laryngeal sonograpphy to evaluate pre and post-operatively
the function of the vocal cords; the bio-humoral evaluation of the parathyroid function and clinical follow up.                                     
Interesting is the standardization of procedures in five steps.            A final striking aspect is the percentage of specific complications that are truly contained for the type of surgery. All the complications, however, with the exception of one permanent hypoparathyroidism, were transient
(two other hypoparathyroidisms and two transient vocal cord palsy).
The paper is accompanied by clear and essential tables, significant images and an updated bibliography.

Author Response

Response to Reviewer 1 Comments

Point 1: Very interesting work on a particular, modern, current aspect of thyroid surgery: robotic completion. Such a large number of operating cases of a niche intervention (even if in a relatively and rightly long period: January 2012 - August 2020) underlines the volume of activity and - therefore - the experience and impact on the effectiveness of the working group.

The study, preparation and post-operative evaluatiuos of patients follows the most up-to-date schemes of international literature: clinical examination, laryngoscopy or laryngeal sonograpphy to evaluate pre and post-operatively the function of the vocal cords; the bio-humoral evaluation of the parathyroid function and clinical follow up.                                      Interesting is the standardization of procedures in five steps. A final striking aspect is the percentage of specific complications that are truly contained for the type of surgery. All the complications, however, with the exception of one permanent hypoparathyroidism, were transient (two other hypoparathyroidisms and two transient vocal cord palsy). The paper is accompanied by clear and essential tables, significant images and an updated bibliography.

Response 1: We appreciate the reviewer for careful and thorough reading of this manuscript and for the thoughtful and positive comments

Reviewer 2 Report

The paper is well written, touches an up-to-date topic of so called minimally invasive thyroid surgery.

In present study the authors analyse the value of the modern technique rpbotic BABA in completion thyroidectomy.

The authors prooved this technique to be feasible and effective in completion thyroidectomy.

The rewiever lacks information about the intraoperative neuromonitoring of the laryngeal nerve. Is it possible in the analysed technique and was it perform?

Author Response

Response to Reviewer 2 Comments

Point 1: The paper is well written, touches an up-to-date topic of so called minimally invasive thyroid surgery. In present study the authors analyse the value of the modern technique robotic BABA in completion thyroidectomy. The authors prooved this technique to be feasible and effective in completion thyroidectomy. The rewiever lacks information about the intraoperative neuromonitoring of the laryngeal nerve. Is it possible in the analysed technique and was it perform?

Response 1: We would like to appreciate your meaningful comment for intraoperative neuromonitoring (IONM) during robotic thyroid surgery. Indeed, it is possible to use the intraoperative neuromonitoring system in bilateral axillo-breast approach(BABA) robotic thyroid surgery and we have used the intraoperative neuromonitoring of the laryngeal nerve for all the cases of robotic thyroid surgery. The NIM Response 3.0 System (Medtronic Xomed,Jacksonville, FL) was used for IONM of recurrent laryngeal nerve. At first, at our hospital, the stimulation probe was inserted through the right axillary incision after flap dissection for BABA robotic thyroid surgery. Since 2016, a nerve stimulator has been connected to the robotic instruments of the daVinci robotic surgical system, so intraoperative nerve monitoring became feasible without inserting an additional stimulation probe.

So, we have added the following brief description of the intraoperative neuromonitoring system in this study.

The surgical techniques of BABA robotic thyroidectomy have been described in detail elsewhere [4]. A total of 33 patients used the intraoperative neuromonitoring system during BABA RCT to facilitate identification of the recurrent laryngeal nerve and prevent nerve injury. (Page 2, 2. Materials and Methods section, line 9)

Figure 1. Adopted from

Bae, D.S.; Kim, S.J. Intraoperative neuromonitoring of the recurrent laryngeal nerve in robotic thyroid surgery. Surg Laparosc Endosc Percutan Tech 2015, 25, 23-26, doi:10.1097/SLE.0000000000000074.

Reviewer 3 Report

The paper is well written and adresses an interesting topic.
One interesting conclusion is drawn: the feasibility of completion thyroidectomy via the same robotic approach as the initial surgery.
In addition, the study outlines well, that the optimal surgical timing of completion thyroidectomy remains unclear.

Line 156: "The cosmetic results after BABA RCT were excellent" - please specify how the cosmetic results were evaluated.

Please specify which surgical robots by Intuitive Surgical were used.

Author Response

Response to Reviewer 3 Comments

Point 1: The paper is well written and adresses an interesting topic.

One interesting conclusion is drawn: the feasibility of completion thyroidectomy via the same robotic approach as the initial surgery.

In addition, the study outlines well, that the optimal surgical timing of completion thyroidectomy remains unclear.

Line 156: "The cosmetic results after BABA RCT were excellent" - please specify how the cosmetic results were evaluated.

Response 1: We appreciate your insightful comment. For 33 patients, there was no flap injury and wound problem through the review of medical records. However, we did not analyze the cosmetic results with objective methods. Therefore, we thought we had better delete the sentence about the cosmetic outcomes; The cosmetic results after BABA RCT were excellent

Point 2: Please specify which surgical robots by Intuitive Surgical were used.

Response 2: We agree with the reviewer's advice and added information about the robotic system we used. We have added the following brief description of the robotic surgical system in the method section of the manuscript.

BABA RCTs were performed using the da Vinci surgical system: the da Vinci S (from 2008 to 2010), da Vinci Si (from 2010 to 2017), and da Vinci Xi (since 2017) (Intuitive Surgical, Sunnyvale, CA). (Page 2, 2. Materials and Methods section, line 6)

Reviewer 4 Report

This is a well written study of the group’s experience of performing completion thyroidectomy via the same robotic bilateral axillo-breast approach as performed for the initial lobectomies. The study analyses the timing of five steps of surgery of completion versus initial lobectomy and outcomes and conclude that completion thyroidectomy was shorter and had similar complication rates. I have a few questions below.

  1. Several patients underwent lobectomy and central neck dissection without immediate completion thyroidectomy. Could the authors comment on their practice to perform lobectomy, and central neck dissection without immediate completion thyroidectomy and what were the indications to perform later completion thyroidectomy in these patients?
  2. Almost half the patients had papillary thyroid microcarcinoma on final pathology of the initial lobectomy. Could the authors comment on their reasons to perform completion thyroidectomy in these patients?
  3. What was the indication for completion thyroidectomy for the minimally invasive FTC cases?
  4. Was central neck dissection performed for visibly abnormal lymph nodes at the time of surgery or were these diagnosed on preoperative imaging? Were frozen sections used for the nodes at the time of surgery?
  5. The complication of hematoma was treated with evacuation. Could the authors comment if this was done via an open transcervical incision?
  6. Drain removal is mentioned in the results in two patients who had seromas following drain removal. Did all the patients have drain insertion at the time of surgery?
  7. What were the pathologies from the completion thyroidectomies?
  8. What percentage of the patients required or underwent RAI?
  9. Did the patients receive prophylactic calcium and/or vitamin D treatment?

Author Response

Response to Reviewer 4 Comments

Point 1: Several patients underwent lobectomy and central neck dissection without immediate completion thyroidectomy. Could the authors comment on their practice to perform lobectomy, and central neck dissection without immediate completion thyroidectomy and what were the indications to perform later completion thyroidectomy in these patients?

Response 1: We would like to appreciate your meaningful comment on the timing of the completion thyroidectomy. Several studies have been reported about the optimal timing of completion thyroidectomy, but it is still controversial. At our institution, the final histopathologic results are generally reported at least 7 to 10 days after the operation. Because the patients who underwent BABA robotic completion thyroidectomy are usually discharged from the hospital within 3 to 4 days after the surgery, the decision of performing completion thyroidectomy is often determined at the first follow-up visit (2 weeks after discharge). Therefore, we perform the completion thyroidectomy at least 3 months after the initial operation.

Point 2: Almost half the patients had papillary thyroid microcarcinoma on final pathology of the initial lobectomy. Could the authors comment on their reasons to perform completion thyroidectomy in these patients?

Response 2: At our institution, we routinely performed prophylactic central neck dissection for all papillary thyroid cancer patients. If the final histopathologic results were categorized as intermediate, or high-risk group according to ATA guideline, we performed completion thyroidectomy for these patients. In our draft, we described indications for completion thyroidectomy (Page 6, Line 51); At our institution, completion thyroidectomies have been considered for differentiated thyroid carcinoma patients who are categorized as intermediate or high risk based on American Thyroid Association risk stratification system [11].

Point 3: What was the indication for completion thyroidectomy for the minimally invasive FTC cases?

Response 3: Among 3 patients who were diagnosed with minimally invasive follicular thyroid cancer, 2 patients had a large-sized tumor (5.5cm, 6.1cm respectively) and the other patient had rapidly growing indeterminate thyroid nodule at the remnant thyroid gland.

Point 4: Was central neck dissection performed for visibly abnormal lymph nodes at the time of surgery or were these diagnosed on preoperative imaging? Were frozen sections used for the nodes at the time of surgery?

Response 4: If the results of preoperative FNAC were consistent with PTC or suspicious PTC, we routinely performed prophylactic central neck dissection. Frozen sections were used to evaluate the lymph nodes at the time of surgery, if there were suspicious lymph nodes.

Point 5: The complication of hematoma was treated with evacuation. Could the authors comment if this was done via an open transcervical incision?

Response 5: This patient underwent hematoma evacuation through the endoscopic approach using BABA incision without open transcervical incision.

(We have added the above brief description of the hematoma evacuation in the results section of the manuscript, Page 5, 3-3. Postoperative Complications, Line 4)

Point 6: Drain removal is mentioned in the results in two patients who had seromas following drain removal. Did all the patients have drain insertion at the time of surgery?

Response 6: Jackson-Pratt drain was routinely inserted through the left axilla incision after BABA robotic thyroid surgery and removed before discharge unless there was a large amount of fluid.

(We have added the above brief description of the drain insertion in the method section of the manuscript, Page 2, 2. Materials and Methods, Line 14)

Point 7: What were the pathologies from the completion thyroidectomies?

Response 7: In the histopathologic results after BABA RCT, most of the patients had no tumor (23/33, 69.7%), 8 patients had papillary thyroid microcarcinoma (8/33, 24.2%),1 patient had nodular hyperplasia (1/33, 3.0%), and 1 patient had miFTC (1/33, 3.0%).

(We have added the above brief description of the histopathologic results of completion thyroidectomy in the results section of the manuscript, Page 4, 3.1. The patients’ baseline Characteristics and Histopathologic Results, Line 8)

Point 8: What percentage of the patients required or underwent RAI?

Response 8: After BABA RCT, 28 patients (84.8%) underwent radioactive iodine treatment.

Point 9: Did the patients receive prophylactic calcium and/or vitamin D treatment?

Response 9: When the patients showed hypocalcemia after the operation, these patients are administered calcium and vitamin D supplement treatment.